# Conflicts with Wolves Can Originate from Their Parent Packs

**DOI:** 10.3390/ani11061801

**Published:** 2021-06-16

**Authors:** Diederik van Liere, Nataša Siard, Pim Martens, Dušanka Jordan

**Affiliations:** 1Institute for Coexistence with Wildlife, Heuvelweg 7, 7218 BD Almen, The Netherlands; 2Department of Animal Science, Biotechnical Faculty, University of Ljubljana, Groblje 3, 1230 Domžale, Slovenia; natasa.siard@bf.uni-lj.si (N.S.); dusanka.jordan@bf.uni-lj.si (D.J.); 3Maastricht Sustainability Institute, Maastricht University, P.O. Box 616, 6200 MD Maastricht, The Netherlands; p.martens@maastrichtuniversity.nl

**Keywords:** human–animal conflict, wolf behavior, migrating wolves, sheep killing, early-life experiences, bold wolves, learning, depredation, deterrence

## Abstract

**Simple Summary:**

Conflicts with wolves arise because wolves kill farm animals, especially sheep, or approach humans. It is expected that young wolves learn from their parent pack (PP) what their prey is and if it is safe to be near humans. To confirm this, we researched whether the behavior of young migrating wolves (loners), after they leave the pack, resembles PP behavior. Fourteen loners entering the Netherlands between 2015 and 2019 could be identified and genetically linked to their PPs. Loner and PP behavior was similar in 10 out of 14 cases. Like their PPs, some young wolves killed sheep and were near humans, others killed sheep and did not approach humans, while two loners were unproblematic, they did not kill sheep nor were they in proximity to humans. Thus, the PP behavior did predict loner’s behavior and conflicts may be similar between young wolves and their PPs. However, conflicts need not arise. To achieve that, new prevention methods are proposed to teach wolves in the PP not to approach sheep and humans. As a result, new generations may not be problematic when leaving the PP.

**Abstract:**

Transmission of experience about prey and habitat supports the survival of next generation of wolves. Thus, the parent pack (PP) can affect whether young migrating wolves (loners) kill farm animals or choose to be in human environments, which generates human–wolf conflicts. Therefore, we researched whether the behavior of loners resembles PP behavior. After being extinct, 22 loners had entered the Netherlands between 2015 and 2019. Among them, 14 could be DNA-identified and linked with their PPs in Germany. Some loners were siblings. We assessed the behavior of each individual and PP through a structured Google search. PP behavior was determined for the loner’s rearing period. Similarity between loner and PP behavior was significant (*p* = 0.022) and applied to 10 of 14 cases: like their PPs, three loners killed sheep and were near humans, five killed sheep and did not approach humans, while two loners were unproblematic, they did not kill sheep, nor were they near humans. Siblings behaved similarly. Thus, sheep killing and proximity to humans may develop during early-life experiences in the PP. However, by negative reinforcement that can be prevented. New methods are suggested to achieve that. As a result, new generations may not be problematic when leaving PPs.

## 1. Introduction

Wolves in Europe live in territorial packs of about 2–7 animals. A pack resembles a family and usually consists of a breeding pair, their offspring from previous years, and sometimes unrelated wolves. All members of the family pack cooperate in raising the cubs, i.e., they protect them, feed them [1], and train them in social, hunting, and survival skills [2]. Young wolves remain in the pack for at least the first 10 months, but usually disperse between 1 and 2 years of age [3] to find mates or a territory of their own [1,4]. They are then referred to as loners. Compared to other social canids, wolves are parented for a relatively long time [5]. The cubs suckle milk and feed on partially digested food regurgitated by the parents or elder siblings until weaning at 10 weeks of age. During these weeks, they receive more and more carcass pieces and complete, but small, opened carcasses [6]. In this way, they become accustomed to the smell and taste of the prey that constitutes their diet [7]. Food can be not only parts of dead animals, but also plant matter such as berries and mushrooms [8]. Young wolves tend to imitate pack members [9] and around the time of weaning, they begin to follow pack members to carcasses [4,6]. Young wolves observe how prey is killed, and as they become stronger, they actively participate and train to approach and kill prey [7,10]. Indeed, Radinger [11] observed that early in life, wolves learn from their parents and elder siblings what kind of prey to select and kill. Thus, young wolves develop a keen knowledge of what prey is, as well as habits and search patterns that increase their hunting efficiency [1,12,13] and their chance of survival [14]. The development of prey perception, selection, and its coupling with the development of prey catching skills applies to mammalian predators in general [15,16]. It is also generally accepted that the transmission of experience about prey and habitat is instrumental in preparing a new generation. Consequently, the new generation prefers a habitat comparable to the one in which it was reared. Thus, the vertical transmission of experience enables the new generation to survive efficiently [4,17,18,19,20].

Wolves living near people are described as cautious about where and when to travel [21] and tend to retreat when they detect approaching humans [22]. However, under certain circumstances, wolves can lose their fear of humans [23,24]. To prevent aggression towards humans, it is important to keep wolves in the wild and maintain their fear [23]. According to Karlsson et al. [22], wildlife managers should detect wolves that behave boldly toward humans during the early stages of habituation. However, they state that the development of boldness is poorly studied. Dispersing wolves would prefer a habitat, depending on early life experiences [25,26], as is known for coyotes [27]. Therefore, a young wolf’s preference for human habitat may also have evolved during the rearing period, especially if the pack is not fearful of humans. In addition, attraction to humans or human habitat can be based on positive reinforcement by food, dogs, or play opportunities [22,24].

The last Dutch wolf was killed in 1845 [28], but after this extinction, since March 2015, numbers of loners have entered the Netherlands. The habitat they have entered consists of densely populated delta lowland including a dense transport infrastructure, intensive agriculture and urban expansion, and fragmented nature areas. It covers 41.5 thousand km^2^: built-up area and roads cover 15%, forests and open natural areas cover 12%, 19% is water, and 54% is agricultural land [29]. Potential wild prey such as roe deer is present almost everywhere in the Netherlands, but with lower distribution in the coastal provinces. In 2015 the number of roe deer in the Netherlands would be up to 100 thousand [30]. As agricultural areas (excluding glasshouse area) and natural areas (excluding waters) sum to 27.2 thousand km^2^ [29], the density is estimated to be 3.7 roe deer per km^2^. The other Dutch wild ungulates (red deer, fallow deer, and wild boar) only live in restricted, fenced areas, the largest being the Veluwe (about 1000 km^2^).

The recent observations of loners have been along roads, in agricultural fields and in villages, or identified when found as road kills, or from DNA samples, e.g., from kills of farm animals. Identification of individual wolves is performed in a current wolf DNA sampling program by the Central European wolf (CEwolf) consortium [31], a coalition of Dutch, Belgian, and German scientists, amongst others. This program also makes it possible to identify their parent packs. Therefore, there is a unique opportunity to test the hypothesis that habits in prey and habitat choice are similar between parent packs and loners. That is the aim of this paper. If there is a high degree of similarity, this study also has practical implications. Sheep were the only livestock species attacked in the Netherlands in the period 2015–2019 [32] and (together with goats) the most preyed livestock species by wolves across Europe [33]. Therefore, whether a parent pack kills sheep may predict what a loner would prefer and predict whether the loner would become problematic towards sheep. This prediction differs from the common assumption that sheep are easy prey [8,34,35,36]. Similarly, whether loners will be in proximity to a human environment may also relate to a habit of their parent packs.

## 2. Materials and Methods

Twenty-two wolves had entered the Netherlands as loners between March 2015 and March 2019. Of these, 17 were identified by DNA analysis and these are the subject of our study. As wolves are new in the Netherlands, the sight of a wolf, or the killing of farm animals was closely followed by the media, dedicated websites, and general public. For DNA analysis, samples of feces or hair were collected from a site where a live wolf was seen, or could be found, without observation of the actual wolf. Slime was collected from wounds of killed farm animals. Tissue from dead wolves was also sampled. These samples had to be collected soon after the source was discovered because DNA quality can decline rapidly under field conditions and cannot be reliably analyzed after two days [37]. DNA analysis of these samples was performed and published by the teams of Animal Ecology of Wageningen University, the Netherlands, Research Institute for Nature and Forest, Geraardsbergen, Belgium and Naturschutzgenetik of the Senckenberg Forschungsinstitut, Gelnhausen, Germany. These teams have standardized their DNA analysis [38] and are collaborating in the CEwolf consortium. They identify wolves, i.e., assign GW (grey wolf) numbers, and register individual wolves and reproducing packs. Genotyping is described by Harms et al. [37] and the Senckenberg Forschungsinstitut [39]. Control-region sequences of mitochondrial DNA are analyzed to verify if it is a wolf species. If so, identification of each individual is possible by additional microsatellite and single nucleotide polymorphism (SNP) genotyping of biparentally inherited and patrilineal Y-chromosome markers [38,40]. The success rate of SNP genotyping is 87% [41].

The behavior of each identified loner was classified as (1) known to kill sheep or not, within 6 months after the loner was detected in the Netherlands, and (2) known to be in proximity to humans or not, also within 6 months after the loner was detected. We chose the 6-month period, because we were interested in the choices of young wolves that are relatively inexperienced, not territorial, and not yet developing their own location-specific routines, and while migrating, probably mainly relying on experiences gained in the parent pack. If the loner was involved in a farm animal attack this was tabled in freely available Excel files on the website of the Dutch interprovincial BIJ12 department [32]. Kills of Dutch farm animals are always reported to the BIJ12 department, as this department financially compensates the losses if wolf DNA was found in wound swabs. The class “being in proximity to humans” was applicable whenever there were eyewitness descriptions, photos, or videos indicating that a wolf had entered a human settlement (village, town, etc.), or eyewitness descriptions, photos, or videos indicating that a wolf had approached humans to a distance of 30 m or less, despite obvious opportunities to stay at a greater distance. The 30 m distance is defined by the Large Carnivore Initiative for Europe LCIE [24] as a close encounter between humans and wolves. In relation to proximity, loner identities were determined based on the Wageningen Animal Ecology team’s identification as reported in news publications. If a loner was only sighted, but its DNA identity was not provided in the reference, we assumed it was the DNA-identified wolf whose DNA track was found in or within 15 km of the sighting. This was possible because a trail of subsequent nearby wolf sightings appeared in the media suggesting that it was the same individual. The trail could match a repeated DNA sample that verified the individual in question. The identity of a sighted wolf could not be estimated if no DNA evidence was collected within 15 km of the sighting. We chose the 15 km distance, as loners may pause migration and temporarily stay in a certain area [42]. We considered this to be similar to a stay in a territory. The 15 km distance covers the distances between the start and end sites of a travel day of Polish wolves in their territory (max. 11 km [12]), as well as the maximum total daily distance between six site measurements per day of German wolves in their territory (max. 13 km [42]).

Parent pack recognition was possible by microsatellite and SNP genotyping allowing the detection of alleles stemming from both the mother and the father [37,39,43]. This genotyping could match with the CEwolf consortium DNA profile database of known wolf individuals and packs and enabled the consortium to trace parent packs of the Dutch loners. Where known, parent packs of loners involved in farm animal attacks were listed in the freely available BIJ12 Excel file [32]. We used the one published 4 January 2020. For loners not associated with farm animal kills, parent packs were disclosed by the Wageningen Animal Ecology team in their publications on loners.

To find descriptions of the parent pack’s behavior in relation to sheep kill and proximity to humans during rearing of the loners, the birth year of the loner was estimated from autopsy reports [44,45,46], reports of tagged individuals caught while still with the parents [47,48], or when the individual was first detected within the pack through DNA analysis [49,50]. These data were combined with the first year of reproduction of the parent pack [49], as well as the assumption that May is the month of birth [12]. If a first DNA detection of a loner was revealed while still in the parent pack that had been reproducing for several years and was frequently monitored by DNA sampling, it was determined that this loner was born in May prior to the detection. 

Descriptions of the selection of sheep as prey or proximity to humans by the parent pack of the loners were found via Google search. Selection of sheep as prey was also checked in government databases of farm animal kills [51,52,53]. A 10-month period was chosen for descriptions of parent packs: between 1 May of the loner’s birth year and 1 March of the following year. Births take place in May [12], so the mentioned period covers the usual minimum period during which young wolves remain in the pack [3]. During this period the loner is raised and gets its first experiences regarding prey and possible proximity to humans by the pack [4]. Since all loners that entered the Netherlands stemmed from German packs, we used the German words for pack (Rudel), farm animal damage (Nutztierschäden), farm animal attack (Nutztierrisse), sheep killed (Schafe getötet), near people (Nähe Menschen), and not being shy (keine Scheu) as keywords. Thus, the following search strings were used: “Rudel” and “location parents” and “year of birth” (for example “Rudel Ueckermünder Heide 2014”). We also used “Rudel” and “location parents” and “Nutztierschäden” and “year of birth”, “Rudel” and “location parents” and “Nutztierrisse” and“year of birth”, “Rudel” and “location parents” and “Schafe getötet” and “year of birth”, “Rudel” and “location parents”, and “Nähe Menschen” and “year of birth”, “Rudel” and “location parents” and “keine Scheu” and “year of birth”. We additionally combined these keyword sequences with the GW-number of the involved loner or the GW-numbers of the respective parents. All of the first 20 hits were read and filtered with respect to evidence of sheep killing, entering human settlements, or being within 30 m of humans. If a filtered hit contained the location of a sheep kill or a wolf observation, the parent pack involved was classified as “known to kill sheep” or “known to be in proximity to humans”. The pack was identified either by DNA or by the distance between the location of the sheep kill or observation, and the center of that pack’s territory, if DNA identification was not revealed in the references. The location of pack territory was provided by the Dokumentations- und Beratungsstelle des Bundes zum Thema Wolf DBBW [54] in a map with territory locations as circles with a scaled radius of 8 km. Although territory size varies between and within packs depending on season and prey abundance [12,42], the area of repeated presence of German pack members was determined by DNA analyses and/or repeated independent sightings by wildlife experts [55]. The German authorities then translated these results into the aforementioned published circles as indications of where the packs were located. The center of this circle was used to measure the distance between the parent pack and the location of a kill or observation. A range of 15 km was chosen for the same reasons as described above for the loners. Within this range, it was assumed that the kill or observation involved wolves from the pack. If there was more than one pack near the kill or observation site, it was assumed that the kill or observation involved wolves from the nearest pack. If Google searches on a parent pack within the period of May–March, which begins in the loner’s birth year, did not reveal any evidence of sheep killing or proximity to humans, the parent pack was classified as “not known to kill sheep” or “not known to be in proximity to humans”, respectively. For some loners the birth year could remain uncertain and could be one or two years before the first discovery in the Netherlands. If this was the case, we looked for descriptions of prey choice and proximity by members of the parent pack for the period between 1 May and 1 March in both possible birth years. During these periods, any description of sheep kill or proximity that met the above criteria was used to classify the parent packs of these loners. Only when no descriptions were found were they classified as “not known to kill sheep” or “not known to be in proximity to humans”.

For the parent pack and the loners four classifications could apply: (1) no kill of sheep, no proximity to humans, (2) no kill of sheep, proximity to humans, (3) kill of sheep, no proximity to humans, and (4) kill of sheep, proximity to humans. The classification of the parent pack as independent variable and that of the loner as dependent variable was analyzed in a multinomial logistic regression. In addition, both classifications were cross-tabulated and Cramér’s V coefficient for their association [56] was calculated. IBM-SPSS statistics version 25 was used for these calculations.

## 3. Results

Between March 2015 and March 2019, 17 non-settled loners entering the Netherlands were detected and identified (Table 1). The median interval of entry of successive loners into the Netherlands was 1.5 months (min: 0 and max: 18; Table 1). Only once were two different loners detected in the Netherlands for the first time on the same day. The distance between the locations of these first detections was 62 km. For 14 of these loners, a parent pack could also be traced back, which was always in Germany. There were 11 parent packs involved: seven were in Niedersachsen, two in Mecklenburg-Vorpommern, one in Sachsen, and one in Brandenburg (Table 1). Three pairs of loners were related: two females stemmed from the Ueckermünder Heide (nr 2 and 17); nr 2 was observed in September 2016, nr 17 in March 2019. Two males were from the Barnstorfer Moor pack (nr 6 and 7). They were observed in February and March 2018. One male and one female were from Babben-Wanninchen (nr 4 and 11). The male was first observed in October 2017 and the female in May 2018. The year of birth could be reliably estimated for 11 loners (nr 1, 3, 4, 5, 6, 7, 11, 13, 15, 16, 17). The median age of these loners was 20 months (min: 9; max: 30; Table 1).

Fourteen loners were responsible for attacks on sheep between March 2015 and March 2019 (Table 2); 203 kills were recorded [32]. Table 2 summarizes references that describe the loners killing sheep and proximity to humans, or not.

Three of the 17 loners (18%) were not associated with killing sheep in a 6-month period following their detection (wolf nr 2, 3, and 13, Table 2). Wolf nr 2 was photographed from a distance in September 2016 defecating in an open farmland at daylight [77]. DNA was taken from feces collected at that spot [49,58]. DBBW [49] reported that this female was from the German pack Ueckermünder Heide in Mecklenburg-Vorpommern. She had also been detected in Engden, Germany, in August 2016 [78]. After the Dutch encounter, she was never detected again. Wolf nr 3 was killed by a car in March 2017. His DNA was also never linked to the killing of farm animals. The autopsy revealed that his stomach contained remains of hare [46]. Wolf nr 13 was only detected by feces samples collected in August 2018 in the Midden Veluwe nature area. It turned out that she settled there, as her DNA was repeatedly found thereafter in this area for 6 months in a row. 

Three identified loners (nr 1, siblings 6 and 7) were observed in proximity to humans (near people and walking through villages). They also killed sheep (Table 2) and formed together 21% of all identified wolves that killed sheep. Wolf nr 1 was recorded traveling through Niedersachsen in Germany and near people since February 2015 [50] and entered the Netherlands in March 2015. He was filmed walking along the footpath of the main road through the Dutch village of Kolham [67] (Figure 1), on several occasions walking alongside cars, crossing fields and roads near people and cars in broad daylight.

Wolf nr 6 was recorded walking through the center of several villages such as in Bennekom where on 21 February 2018 this wolf was filmed twice [71] and subsequently filmed the same night in Veenendaal [79] at 9 km distance (Figure 2), passing cyclists and cars. This wolf was seen both in broad daylight and at night.

His brother, wolf nr 7, was repeatedly seen in Belgian villages Dilsen-Stokkem and Meeswijk [80,81]. All other loners were never reported to be seen in proximity to humans.

The behavior of the parent pack in relation to killing sheep and proximity to humans at the time that a loner was born is described in the references in Table 3. No references of sheep killing were found for two parent packs (relating to loner nr 10 and 13). For the other nine parent packs there are references of sheep killing. DNA proof on killed sheep applied to five of these nine (relating to loner nr 3, brothers 6 and 7, and loners 8, 15, 16). For the remaining four parent packs (relating to loner nr 1, sisters 2 and 17, siblings 4 and 11, and loner nr 5), references related to the distance between the site of a sheep kill and the center of the parent pack territory were used. However, the sheep killing results of two of these parent packs were not consistent in the course of time. This applied to the parent pack Ueckermünder Heide (relating to loner nr 2 and 17) and the parent pack Babben Wanninchen (relating to loner nr 4 and 11). The parent pack Ueckermünder Heide started to reproduce in 2014; loner nr 2 was born either in 2014 or 2015 (Table 1). No sheep kills could be associated with this pack in these years (Table 3), whereas in 2017, the birth year of loner nr 17 (Table 1), sheep kills occurred within 15 km of the territorial center of this pack [82,83]. The parent pack Babben Wanninchen started to reproduce in 2013. Loner nr 4 was born in 2015. The only kill of sheep during his rearing period from May 2015 to March 2016 occurred in January 2016 [84], but this was 5 km from the center of a neighboring pack and 13 km from the Babben Wanninchen parent pack. Therefore, this kill was not attributed to the Babben Wanninchen pack (Table 3). As of 2016, there was another father. Loner nr 11 was born in 2016 (Table 1). In October 2016, there were two attacks on sheep within 6 km of the center of the nearest pack, namely the Babben Wanninchen pack [84]. Thus, here we attributed sheep kill to this parent pack (Table 3).

The parent pack of loner nr 1, born in 2014 at the Truppenübungsplatz, Munster, Niedersachsen (Table 1), was actually not related to references describing the killing of farmed sheep in 2014/2015. Nevertheless, we assumed an involvement of the pack in the killing of sheep (Table 3), because the pack is located in the Lünerburger Heide. This heather habitat is well known for the free-ranging herds of a sheep breed called Heidschnucke [92]. It is likely that the parent pack has experienced these sheep. Actual wolf kills of such sheep in this area must have been occurring, as this motivated shepherds to install night enclosures in 2015. A lack of reference in the database of farm kills in Niedersachsen can be explained as reporting and DNA sampling of killed free-ranging Heidschnucke was not considered useful and was not performed [93]. An additional argument is that the pack was known to kill sheep in previous years [85]. 

References of proximity to humans were found for three of the 11 parent packs (of loner nr 1, brothers 6 and 7, and loner nr 8). These involved sites of proximity within 15 km of the center of the parent pack’s territory. In the case of loner nr 1, his parents and siblings of Truppenübungsplatz, Munster pack were well known for their proximity to humans [50,69]. Moreover, a sibling born in 2015, that remained in Germany, ID-numbered as GW369m, and was not one of the Dutch loners, was fitted with a VHF-transmitter around his neck. This wolf also repeatedly followed humans and attacked dogs that were being walked. For this reason, he was shot [50]. In the second case, the Barnstorfer Moor parent pack of sibling loners nr 6 and 7, eyewitness reports mention wolves from this pack being in proximity [90]. In the third case, the Schneverdingen parent pack of loner nr 8, a video from February 2016 shows a wolf approaching a waiting car. This happened at a distance of 3 km from the center of their territory [91].

The characteristics of the parent packs at the time the loners were reared, and the characteristics of the loners themselves have been summarized and cross-tabulated in Table 4. These wolves only classified in the categories (1) no kill of sheep, no proximity to humans, (3) kill, no proximity, and (4) kill and proximity. No parent pack or loner was characterized by the class (2) no kill, proximity. The multinomial logistic regression model was significant (*p* = 0.02) with behavior category of the parent packs as predictor variable. Moreover, the Cramér’s V coefficient was 0.64, showing a significant association between behavioral characteristics of the parent pack and of their loner (*p* = 0.02).

## 4. Discussion

Though based on a limited sample size, our results are in an agreement with the hypothesis that there is a high degree of similarity in prey and habitat choice between parent packs and their loners. They are similar regarding sheep killing or not and being in proximity to humans or not.

DNA recognition allows reliable documentation of individual loners. However, not all observations in our study are backed up by DNA identification, which could raise doubt whether some data are correct. For example, for wolf 6, DNA samples were not taken at each confrontation. Therefore, it can be questioned whether it was always the same individual. Nevertheless, it is very unlikely that different individuals were present at the same time and place on the subsequent days of the observations. First, there were no resident wolves in the Netherlands until February 2019 [94]. Second, it is unlikely that two loners migrated on the same track and at the same time, as the median time interval between successive loners entering the Netherlands was 1.5 months. Even in the single case of a first detection on the same day, the distance between loners was 62 km (loner 8 and 9, Table 1).

Another inaccuracy could be seen in the assessment of the parent pack involvement when DNA identification is lacking and the distance between the site of a sheep kill or proximity encounter and the center of the parent pack’s territory is used instead. The shape and size of the territory at the time of the kill or proximity encounter was unknown, the 15 km limit may not cover the actual exploration range of pack members, and members from neighboring pack territories may explore the parent pack area [42]. Thus, the possibility remains that a wolf involved was other than a parent pack one. Nevertheless, Google searches were a valuable tool to classify sheep kills or proximity to humans. We give some examples to illustrate this. For example, Koerner [69] and other sources [50] describe that members of the parent pack of loner 1 Truppenübungsplatz in Munster Nord have been in close proximity to humans repeatedly since 2012 and also early 2015. Another example provided by Proplanta [82] and confirmed by Welt [83] described the Ueckermünde as a regional hotspot for the killing of farm animals, including sheep. This would apply to 2017 and 2018. This hotspot is 5 km from the center of the parent pack Ueckermünde Heide, while loner nr 17 is born in 2017. Such reports of local experiences contribute to the reliability of the classification of the parent pack, despite the lack of DNA proof. 

The repeated wolf reporting shows that the presence of wolves is highly profiled and covered in the media in both Germany and the Netherlands, especially when wolves are near humans or kill farm animals. This is partly due to unfamiliarity with wolves and concern for the safety of people in a region with a new large predator. Wolves have been reproducing in Germany only since 2000 [95,96], while the first proven Dutch wolf presence was in 2015 (wolf nr 1), after more than 170 years of absence [28]. Despite potential errors, the nominal measurements accumulated to a result in agreement with the hypothesis. Improved assessment of which wolf was involved would enhance the reliability. This means to enhance identification through means other than DNA, such as registering visual or behavioral characteristics that describe the individual wolf, similar to the registration of individual wolves in Yellowstone park [97]. Another possibility is to use the howl as means to identify individual wolves [98], although loners tend to howl less than members of a territorial pack [99]. 

Of the 17 identified loners, three loners (nr 2, 3, and 13) were not associated with kills of farm animals (Table 2). Two of them were seen (wolves nr 2 and 3). Thus, the probability of actually observing a loner that could not be detected by farm animal kills was at least 2 out of 17, so 12%. These two wolves could be the tip of an iceberg and representatives of a larger subpopulation of unobserved wolves: wolves that did enter the Netherlands, but did not kill farm animals and went unobserved. If so, it is remarkable that a subpopulation of loners enters and crosses the Netherlands without killing farm animals; sheep in particular, as sheep are considered easy prey [8,34,35,36]. In this respect, there are three significant observations. First, the three wolves mentioned above were present in the same or similar areas where other wolves have killed sheep. Second, the sheep density in the Netherlands is three times higher than in German states where the parent packs live [100]. Therefore, the Dutch conditions make it even more unlikely that a loner does not encounter and kill a sheep. Thirdly, access to sheep has hardly any threshold, as by the absence of wolves over 170 years, Dutch sheep are kept without special protection from wolves. Thus, the availability of sheep cannot explain why there are loners that do not kill sheep. Other researchers also showed that prey preference cannot be fully explained by availability [8,35,101,102,103,104,105,106]. Therefore, prey preference is apparently not merely a matter of stochastic laws. 

Other explanations for selecting sheep as prey are also mentioned, such as development of specialization [101,107,108], or the development of an ecotype with a feeding preference or a feeding habit [108,109]. Thus, learning mechanisms are involved. References to learning mechanisms are “development of hunting experience” and “learning from parents” [10,13,35] or “learning differences between packs” [102]. Indeed, similarity between choices of wolves in the parent pack and their migrating offspring gives a corresponding explanation. This possibly applies to the three wolves mentioned above, if they stuck to prey familiarized to them through experiences gained in the parent pack. The loners were young wolves at detection in the Netherlands with a median age of 20 months, as commonly found for young dispersing wolves [3,42]. If in the period before dispersal, parent packs fed their young wild prey only and showed their maturing young how to find, approach, and kill wild prey (like roe deer), then the loner has left the pack only familiarized with these species. In cases where sheep have not been part of the diet or hunt, they have not learnt about sheep as prey. We suggest that a wolf will have no drive to explore new prey, such as sheep, as long as familiar wild prey density sustains survival and reproduction. The common notion that sheep are easy prey may only be true for wolves that are familiar with sheep. Conversely, the prediction that wolves will not kill sheep if there is sufficient wildlife [35,36,110] may be incorrect for wolves that are familiar with sheep.

Wolf nr 3 is a special case, as this wolf was not linked to the killing of sheep, but his parent pack was. In the reasoning above, this is not expected. Wolf nr 3 was born in 2015 (Table 1). There were no sheep kills in or near his parent pack territory in 2015. However, the first known kill of sheep was in 2016 as DNA samples showed that the parents (GW203f and GW339m) killed sheep together on 24 January 2016 [52]. Wolf nr 3 would then have been 8 months old. It is not clear if at that time he was still present in the pack, as his DNA was never detected at sheep kills in or near his parent pack territory. Thus, it is possible that wolf nr 3 had already left the pack early in January 2016 and had not gained experience in sheep killing in the parent pack. Leaving the pack at a relatively young age in January or February probably was also the case for wolves nr 1, 6, and 7, which were 9 to 10 months old when they reached the Netherlands (Table 1). However, the conclusion in the case of wolf nr 3 is that parent pack and loner behavior in killing sheep is dissimilar. 

In cases where the abundance of familiar wildlife prey is falling, it cannot be ruled out that a naïve wolf starts exploring farm animals like sheep on its own. Roe deer are wild ungulates strongly preferred by German wolves, making up 52–54% of the consumed biomass [34,111]. Therefore, it could be expected that German wolves, familiar with and normally focused on roe deer, tend to start to kill sheep in areas where sheep densities are high and roe deer density falls. As discussed above, the Netherlands has a sheep density three times higher than the German states where the parent packs are living. However, the Dutch roe deer density of 3.7/km^2^ is likely to be lower than densities in the German states where the parent packs are living. For instance, in Sachsen the estimated density was 4.8 in 2015, as calculated from the annual roe deer cull [112], wolf kills [34] and agriculture and nature area size [113]. The relatively high sheep density and likely low roe deer density in the Netherlands make it all the more remarkable that loners 2, 3, and 13 did not start to kill sheep when entering this country. It strengthens the conclusion that prey preference of loners cannot be fully explained by prey availability and that early life experiences in the parent pack are involved. Exploration of new sources is seen with wolf nr 13, which did not kill sheep during migration (Table 2) but did kill sheep once after settlement in February 2019 as a single wolf. This was in October 2019 [32]. It is not known however what triggered this kill. A trigger could be a drop in wildlife abundance at that time or an increase of disturbances of wildlife by tourists or shooters [12].

Three of the 17 identified loners (nr 1, 6, and 7) were in close proximity to humans (Table 2). An explanation could be that the Netherlands is densely populated with 513 people per km^2^ [114], which may result in proximity. However, the wolves travelled through human settlements during daylight and did not try to avoid the human environment (see references in Table 2). Even in the Netherlands, and especially in villages and towns where wolves have been seen travelling, there is ample opportunity to travel through forests or fields adjacent to these human settlements rather than through streets with traffic, pedestrians, and bicycles, like in the cases of wolves nr 1 and 6 (Figure 1 and Figure 2). The simplest explanation would be then that these wolves have already had positive reinforcement and experience with a human environment and chose this over adjacent environment at distance from humans. 

Positive experience with a human environment may have started in the parent pack. Indeed, parent pack members that were known to be in close proximity to humans or human settlements had similarly behaving loners (Table 4). Moreover, similarities in proximity to humans applied to siblings. Both wolf nr 1 and his one-year-younger brother GW369m, which stayed in Germany [50], and the two brothers nr 6 and 7 were in proximity to humans. In addition, if there were no proximity references, then this also applied for both, siblings and parent packs, as in the case of half-brother nr 4 and half-sister nr 11 and sisters nr 2 and 17 (Table 2, Table 3 and Table 4). These findings confirm that besides prey choice, young wolves may learn by following and imitating their parents or siblings [4,9] to be in proximity of people. Already in the parent pack, young wolves may experience that human environments and humans need not be feared and may even be rewarding. The consistency between parent pack and loner choices regarding proximity to the human habitat corresponds to other studies where habitat preference of dispersing wolves is shown to be determined by early life experiences [25,26]. Such vertical transmission of preferences is thought functional for survival [4,17,18,19,20]. However, this transmission implies that human–wolf conflicts may increase not only because of expanding wolf populations, but also by reproduction of parent packs with the habit to be in proximity of humans and a subsequent dispersal of a new generation that also prefers to be in proximity to humans. 

Kuijper et al. [36] recommend discouraging wolves from approaching human habitat by deterring them while maintaining high abundance of wildlife prey. They follow Huber et al. [115] and Newsome et al. [116], who assume that European wolves are habituated to humans, though it is not clear what “habituation” means in these references. Moreover, Kuijper et al. [36] do not refer to rearing conditions of wolves and their consequences for the development of a tendency to approach human habitat. LCIE [24] assumes positive conditioning as part of wolf habituation, through food, dogs, or play opportunities. However, LCIE only recommends deterring of so-called bold wolves. These wolves would repeatedly tolerate or actively approach people within 30 m. The main tool would be to prevent people from feeding wolves. Again, recommendations do not clearly link with the parent pack, though LCIE does recognize maternal and sibling influences on the development and consistency of bold wolf behavior. For unknown reasons LCIE does not mention paternal influences.

Given the literature about vertical transmission of experiences as mentioned before and the results in this study, we presume that prevention of problems may be promoted by discouragement of parent pack wolves approaching human dwellings or humans. A deterring focused at the reproducing pack is also expected to increase the threshold for young inexperienced wolves to follow and copy the choice to enter the human habitat. As a consequence, a tendency to find rest, shelter, and food in environments outside human dwellings may be maintained across generations. Moreover, deterring bold wolves only would be a very late measure. In our view the deterring should take place as soon as the approach happens, so with any wolf that crosses villages or approaches humans, whether fitting the definition of “bold” or not. This is in contrast to the currently accepted policy in Germany, the Netherlands, and Belgium not to take measures when a non-bold wolf enters a village. A spin-off of keeping wolves at a distance from human settlements could also be that wolves are less likely to encounter and subsequently gain experience with domestic animals. 

When weighing the problem of proximity to humans with the problem of killing farm animals, the latter is more severe, as a wolf attack has a severe impact on the welfare of involved sheep, results in production losses [117] and it may lead to (illegal) killing of wolves [35,36,100,118,119]. Moreover, it is a particular problem with loners as their migration pattern is difficult to predict. As a consequence, prevention measures may be taken ad hoc, too late, and only after kills of farm animals have taken place. In addition, parent packs may have learnt to overcome prevention measures, such as electric fences [106], for instance by jumping over them. Young wolves may therefore have also learnt to overcome (electric) fences. So when migrating, such wolves may not be sufficiently thwarted to kill farm animals. The difficulty in predicting the occurrence of problems with migrating wolves implies the need to reduce their likelihood in another way, which can be through developing methods preventing farm animal killing experiences, while the individual wolf is still in the parent pack. Thus, in particular, herds near the pack should be properly protected. However, current methods such as electric fences and guarding dogs are not always effective or possible [106]. Therefore, additional methods need to be developed. We suggest testing the application of a sensor in two sheep in a fenced herd, wirelessly connected to a deterring system that can cover and protect the herd. This sensor would be immediately activated by the stress in a sheep due to the approach of a wolf [120]. A second measure that can teach wolves not to select sheep is a collar for sheep, activated the moment a wolf bites in the neck and then delivering a painful electric shock only to the wolf [121]. A third way is to provide a sheep carcass injected with an emetic, placed near the wolves’ den. When an adult wolf feeds on this cadaver, the nausea is expected to stop the wolf from selecting this carcass and possibly other sheep as food, also in the course of feeding the young. The methods combined contribute to teaching wolves and their offspring not to approach, bite, and consume sheep. This seems effective when alternative food resources are abundant, such as wild ungulates. The problem of (unpredictable) attacks on sheep may therefore be largely solved, if all levels of the wolf’s hunting (the approach, the biting, and the consumption) are negatively reinforced in parent packs. Those packs would thus raise new generations of migrating wolves without prior experience with sheep as prey and not necessarily killing sheep. 

## 5. Conclusions

Though with a limited sample size, this study shows that the behavior of parent packs significantly predicts the behavior of their migrating offspring. Parent packs that killed sheep and were or were not regularly seen near humans or human settlements had similarly behaving loners. In addition, there were loners that behaved unproblematically like their parent packs: they did not kill sheep nor were they near humans. This suggests that the common notion that sheep are easy prey may only be true for wolves that are familiar with sheep. Conversely, the prediction that wolves will not kill sheep if there is sufficient wildlife may be incorrect for wolves that are familiar with sheep. The results confirm that in addition to prey choice, young wolves may learn to be around humans by following and imitating their parents or siblings. Vertical transmission of experience implies that human–wolf conflicts may increase not only due to expanding wolf populations. They may also increase by reproduction of parent packs with a habit of killing sheep or being near humans, and a subsequent dispersal of a new generation with similar habits. The difficulty in predicting the occurrence of problems with migrating wolves implies the need to reduce problems by new methods. These new methods should generate negative experiences of killing farm animals and being in proximity to humans while the individual wolf is still in the parent pack.

## Figures and Tables

**Figure 1 animals-11-01801-f001:**
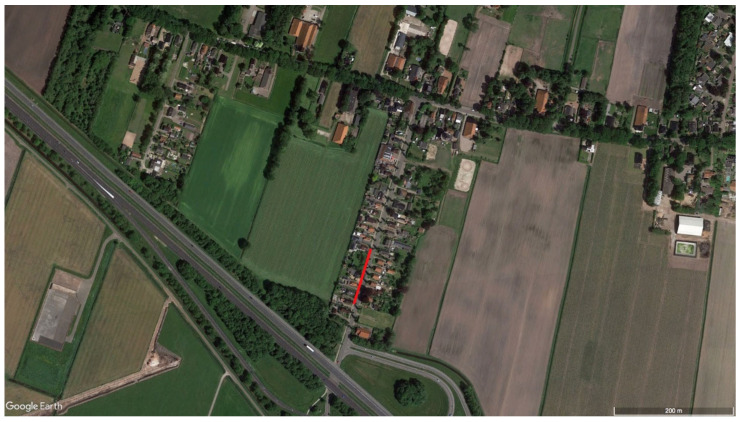
Satellite image overviewing the village Kolham and the trajectory (in red) from south to north of wolf 1 when filmed. Image: ©Google Earth.

**Figure 2 animals-11-01801-f002:**
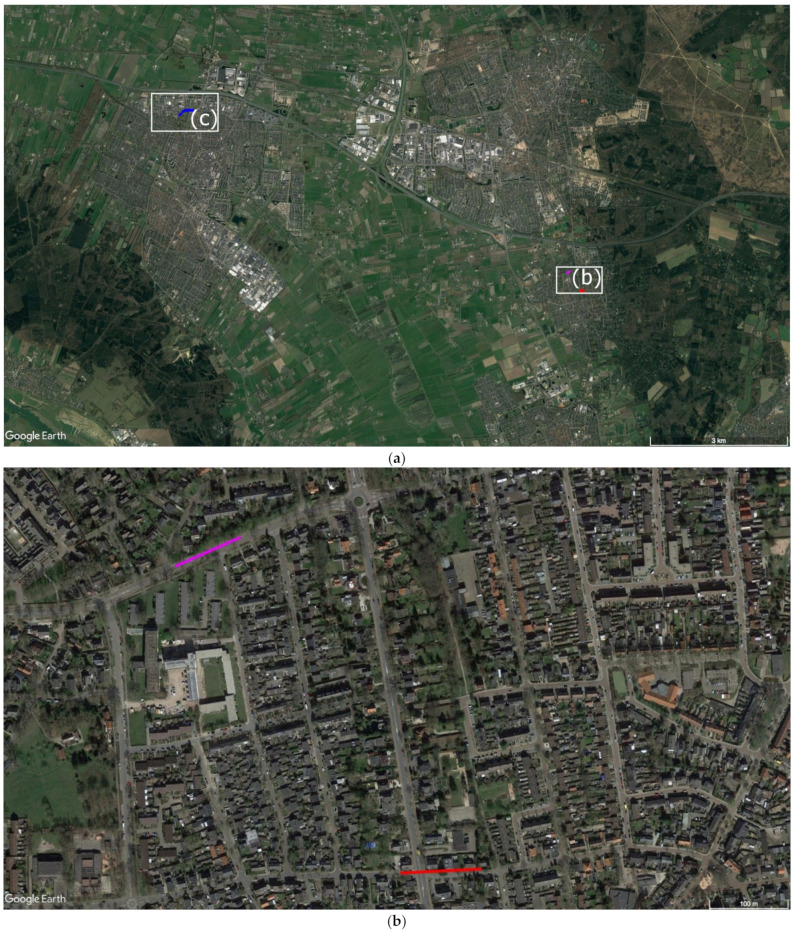
Satellite image overview (**a**) of the subsequent east to west video-recorded locations of wolf 6 in one night: (**b**) in red and pink in Bennekom; (**c**) in blue in Veenendaal. Images: ©Google Earth.

**Table 1 animals-11-01801-t001:** Identified loners that have entered the Netherlands (NL), numbered chronologically with their ID number (nr) based on DNA analyses (f: female; m: male), references of loners including ID numbers, date and location of first detection in the Netherlands, age, year of birth and method of age determination (G: first genetic detection; A: autopsy; C: capture in the pack; F: first litter; without a letter: uncertain) and location of parent pack (N: Niedersachsen; MV: Mecklenburg-Vorpommern; B: Brandenburg; S: Sachsen; n.k.: not known).

Loner Nr	Loner’s ID Nr	References with ID Nr	Date First Detection NL	Location First Observation NL	Age in Months/ Year of Birth	Parent Pack/Land in Germany
1	GW386m	[32,57]	07/03/15	Bargerveen	10/2014G	Truppenübungsplatz in Munster Nord/N
2 ^a^	GW620f	[49,58]	03/09/16	Beuningen	≥16/2014 or 2015	Ueckermünder Heide/MV
3	GW657m	[46]	03/03/17	Veeningen near highway A28	22/2015A	Cuxhaven Langes Moor/N
4 ^b^	GW843m	[32,45]	14/10/17	Laag Zuthem	30/2015A	Babben Wanninchen/B
5	GW680f	[32,47,59,60]	19/12/17	Forest Hardenberg	18/2016C	Lübtheener Heide/MV
6 ^c^	GW955m	[32]	21/02/18	Betuwe area, river Nederrijn	9/2017F	Barnstorfer Moor/N
7 ^c^	GW913m	[32,44]	05/03/18	Lottum- Venlo area	10/2017A	Barnstorfer Moor/N
8	GW954f	[32]	18/03/18	Benneveld	≥10/2016 or 2017	Schneverdingen/N
9	GW953m	[32]	18/03/18	Beckum	n.k.	n.k.
10	GW763f	[32]	04/04/18	Boijl	≥11/2015 or 2016	Daubitz/S
11 ^b^	GW998f	[32,61,62]	03/05/18	Langezwaag	24/2016C	Babben Wanninchen/B
12	GW979m	[32,63,64]	16/06/18	Buitenpost	n.k.	n.k.
13	GW960f	[62]	01/08/18	Midden Veluwe	14/2017G	Göhrde pack/N
14	GWxxxf	[32,62]	03/10/18	Orvelte	n.k.	n.k.
15	GW893m	[32,65]	05/01/19	Damsholte	20/2017G	Eschede/Rheinmetall/N
16	GW965f	[32,66]	22/02/19	Lemelerveld	21/2017G	Die Lucie/N
17 ^a^	GW849f	[32,48]	29/03/19	Hooghalen	21/2017C	Ueckermünder Heide/MV

^a, b, c^: same letters are siblings.

**Table 2 animals-11-01801-t002:** Identified loners and references with evidence of killing sheep and proximity to humans within 6 months after the first detection in the Netherlands.

Loner Nr	Loner’s ID Nr	Killing of Sheep	Proximity
1	GW386m	Yes [32]	Yes [50,67,68,69]
2 ^a^	GW620f	No	No
3	GW657m	No	No
4 ^b^	GW843m	Yes [32,45]	No
5	GW680f	Yes [32,70]	No
6 ^c^	GW955m	Yes [32]	Yes [71,72,73]
7 ^c^	GW913m	Yes [32,74]	Yes [74,75]
8	GW954f	Yes [32]	No
9	GW953m	Yes [32]	No
10	GW763f	Yes [32]	No
11 ^b^	GW998f	Yes [32,61,62]	No
12	GW979m	Yes [32,63]	No
13	GW960f	No	No
14	GWxxxf	Yes [32,76]	No
15	GW893m	Yes [32,65]	No
16	GW965f	Yes [32,66]	No
17 ^a^	GW849f	Yes [32]	No

^a, b, c^: same letters are siblings.

**Table 3 animals-11-01801-t003:** Fourteen identified loners that were genetically linked to a parent pack, with their ID number, and references to their parent pack regarding location (N: Niedersachsen; MV: Mecklenburg-Vorpommern; B: Brandenburg; S: Sachsen), killing of sheep and proximity to humans in the period May of the loner’s birth year (or possible birth years) to March of the next year.

Loner Nr	Loner’s ID Nr	Location/Land in Germany	Killing of Sheep	Proximity
1	GW386m	Truppenübungsplatz, Munster/N	Yes [52,85]	Yes [50,69,86]
2 ^a^	GW620f	Ueckermünder Heide/MV	No	No
3	GW657m	Cuxhaven Langes Moor/N	Yes * [52]	No
4 ^b^	GW843m	Babben Wanninchen/B	No	No
5	GW680f	Lübtheener Heide/MV	Yes [87,88]	No
6 ^c^	GW955m	Barnstorfer Moor/N	Yes * [52,89]	Yes [90]
7 ^c^	GW913m	Barnstorfer Moor/N	Yes * [52,89]	Yes [90]
8	GW954f	Schneverdingen/N	Yes * [52]	Yes [91]
10	GW763f	Daubitz/S	No	No
11 ^b^	GW998f	Babben Wanninchen/B	Yes [84]	No
13	GW960f	Göhrde pack/N	No	No
15	GW893m	Eschede/Rheinmetall/N	Yes * [52]	No
16	GW965f	Die Lucie/N	Yes * [52]	No
17 ^a^	GW849f	Ueckermünder Heide/MV	Yes [82,83]	No

*: DNA analysis demonstrated parent pack was involved in killing sheep; ^a, b, c^: same letters are siblings.

**Table 4 animals-11-01801-t004:** Frequency distribution of sheep kill and proximity to humans characteristics of 14 identified Dutch loners and their parent packs at the time the loners were reared (bold: similar behavior; loner nr in parentheses).

		Loner		
		no sheep kill; no proximity	sheep kill; no proximity	sheep kill; proximity
**Parent pack**	no sheep kill; no proximity	**2 (2, 13)**	2 (4, 10)	
	sheep kill; no proximity	1 (3)	**5 (5, 11, 15, 16, 17)**	
	sheep kill; proximity		1 (8)	**3 (1, 6, 7)**

## Data Availability

The data are contained within the article.

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
