# Peer review of "Conflicts with Wolves Can Originate from Their Parent Packs"

_animals, 2021, doi:10.3390/ani11061801_

Round 1

Reviewer 1 Report

Comments will be emailed to the editor because the comments cannot be copied or attached to the form

Author Response

REPORT 1

  • Include some information on natural prey densities. Since wolves are adaptable, it may be possible that the loners are not regular sheep killers but because of the low natural prey density they may switch to sheep. It usually does not take much for predators to learn to hunt different prey.

We accept what you suggested and made the following changes in the Discussion:

line 485-498: Roe deer are strongly preferred wild ungulates by German wolves, making up 52-54% of the consumed biomass ……

We also added in the Introduction section a description of the Netherlands with elements that relate to the (un)suitability as a wolf habitat. Here we also added descriptions of the ungulate species in the Netherlands that are native wolf prey species in NW Europe.

It can be found at lines 79-92: The last Dutch wolf was killed in 1845 [28], but after this extinction, since March 2015, numbers of loners have entered the Netherlands. The habitat they have entered consists of densely populated delta lowland….

  • I suggest writing more concisely as there is repetition in results and discussion.

We have rewritten several parts of paragraphs of the Discussion and made them more concisely as requested:

line 366

line 384-394

line 410-412

line 420-430

line 446-449

line 453-461

line 466

line 469-478

line 512-517

  • Leave out the Google maps since they do not add value to the article. Everything has been described in detail in the results and discussion. The same applies to the tables.

The comment is correct, the information from Figures 1 and 2 is explained in the Results and Tables (limitedly and in neutral terms), and this information is discussed in the Discussion section. However, we believe there is a significant added value in Figures 1 and 2. As “a picture is worth a thousand words”, the figures clearly show that wolves chose routes right through towns and close to humans, such as main roads, although they had the possibility to use alternative routes through nearby forests or agricultural fields. It is not the location as such that is relevant, but the fact that they may choose areas, packed with traffic, people and houses, which, we believe, is impressive. The pictures add to our point that wolves have not been randomly or by chance entering human settlements, because there is a high density of settlements, but deliberately chose this context in spite of ample alternatives to migrate at distance of humans. That is what we expressed and discussed in paragraph 8 in the Discussion. We therefore suggest to maintain the pictures.

  • In the Summary, 14 loners entered the Netherlands, while in the Abstract 22 loners entered the Netherlands. Which is correct?

In both cases, it is correct.

Summary: 14 loners entering the Netherlands between 2015 and 2019 could be identified and genetically linked to their PPs.

Abstract: After being extinct, 22 loners had entered the Netherlands between 2015 and 2019. Among them, 14 could be DNA-identified and linked with their PPs in Germany.

In Summary and in Abstract 14 wolves that could be identified are mentioned. We decided to add the information in the Abstract, that there were more wolves that entered the Netherlands (that is 22), but not all of them could be DNA-identified (14 out of 22). Therefore, there is no discrepancy between Abstract and Summary.

  • Standardize the font

The font in the submitted paper is uniform. We ask the editor to check if the discrepancy occurred after the submission.

  • Line 50 Rewrite: “but also vegetable matter such as from berries, mushrooms and the like”

We changed this into:

line 50-51: ...also plant matter such as berries, mushrooms and the like

  • Rewrite: “habitat comparable to the rearing one and efficiently survive in it”

We agree that the phrase can be ambiguous. It is not clear what “it” refers to at the end of the phrase. Correction is made in line 60-64.

(Schafen getötet) should be (Schafe getötet)

Corrected.

  • “Nutztierschade” should be (Nutztierschäden), plural or (Nutztierschaden), singular

Corrected.

  • Line 204 and elsewhere … no kill, proximity, (3) kill, no proximity, and (4) kill and proximity.

Specify – no kill of sheep or natural prey?

Kill of what prey?

No proximity to humans or human development? The same applies to proximity.

The classes are explained and defined earlier in Material in Methods line 129-143. To make it clearer, we changed the following starting at line 222: (1) no kill of sheep, no proximity to humans, (2) no kill of sheep, proximity to humans, (3) kill of sheep, no proximity to humans, and (4) kill of sheep, proximity to humans.

However, we wanted to avoid repetition in describing the classes. We therefore suggest that the references to the classes later on in the text remain as they are.

Reviewer 2 Report

Many behavioral characters associated with a given animal species are clearly are innate (e.g., courtship displays) while others may involve social learning from birth to near adulthood. In social mammals parents transmit a variety of characters to their offspring in the form of teaching. Behaviors related to habitat selection, foraging, predation avoidance represent a suite of characteristics that presumably have survival value insofar they persist within the species in question. Wolves are an example of a species that typically live in extended family groups, and where learned behaviors pertaining to foraging behavior, including prey selection and hunting tactics,

In this paper van Liere et al. examine patterns prey selection and habituation to humans of young, solitary wolves that have dispersed from their natal area in relation to the foraging/habituation characteristics of their parental packs. Based on genetic evidence obtained from scat and saliva, they identify solitary wolves which dispersed to the Netherlands to their known parent packs in Germany. The parent pack could be divided up into 4 behavioral categories: 1) wolves which killed sheep and associated with humans, 2) wolves which killed sheep but avoided human contact, 3) wolves which killed native prey, but associated with humans, and 4) wolves which killed native prey, and avoided humans.

Though their total sample size is somewhat limited (14 DNA-identified wolves with known parentage) in light of the 4 behavioral categories examined, they show that prey selection and habituation of young, independent wolves largely reflect the behavior of their family in which they were raised. The majority of the wolves killed sheep, and some associated with humans. However, of the few wolves that did not prey on sheep, none of them associated with humans.

I found this paper to be an interesting contribution from a geographic locale that has been underrepresented in the literature on the current topic. However, I feel that the results could be couched with a bit more caution than articulated in the current version of the manuscript. The authors correctly point out that the identity of the wolf for a certain behavior may have been uncertain. As importantly though, is the limited scope that these data represent. Though the statistical relationship between the behaviors of lone wolves relative to their parent pack was statistically significant, the low number of independent samples calls into question the generality of the results (albeit squaring with similar observations from the literature). Moreover, there are possible temporal issues to consider as well. Wolves learn throughout their lives, not just during their adolescence. I see no reason why the current “rabbit hunters” could not in time become sheep killers. Wolves are opportunistic hunters, clearly capable of prey switching as ecological conditions change. Indeed, the paper falls short in providing the reader with information regarding the diversity and abundance of potential native prey in the ecosystem to which these young wolves dispersed. Thus, a fuller description of the broader ecological context experienced by these dispersing wolves could shed additional light onto the suite of behaviors discussed in this paper.

Author Response

REPORT 2

Many behavioral characters associated with a given animal species are clearly are innate (e.g., courtship displays) while others may involve social learning from birth to near adulthood. In social mammals parents transmit a variety of characters to their offspring in the form of teaching. Behaviors related to habitat selection, foraging, predation avoidance represent a suite of characteristics that presumably have survival value insofar they persist within the species in question. Wolves are an example of a species that typically live in extended family groups, and where learned behaviors pertaining to foraging behavior, including prey selection and hunting tactics,

In this paper van Liere et al. examine patterns prey selection and habituation to humans of young, solitary wolves that have dispersed from their natal area in relation to the foraging/habituation characteristics of their parental packs. Based on genetic evidence obtained from scat and saliva, they identify solitary wolves which dispersed to the Netherlands to their known parent packs in Germany. The parent pack could be divided up into 4 behavioral categories: 1) wolves which killed sheep and associated with humans, 2) wolves which killed sheep but avoided human contact, 3) wolves which killed native prey, but associated with humans, and 4) wolves which killed native prey, and avoided humans.

It is suggested to divide the parent pack in 4 categories. These seem the same categories as we used, but the difference is:

  • that they either are associated with humans or avoided human contact. It is not clear what is meant with association with humans, or human contact. We tried to be as unambiguous as possible and defined proximity in the Material in Methods section (lines 139-143): The class “being in proximity to humans” was applicable whenever there were eyewitness descriptions, photos or videos indicating that a wolf had entered a human settlement (village, town, etc.), or eyewitness descriptions, photos or videos indicating that a wolf had approached humans to a distance of 30 m or less, despite obvious opportunities to stay at a greater distance.
  • that they either killed native prey or sheep. We don´t have the information what kind of prey (food) was chosen by the wolves which didn´t predate sheep. This is not necessarily “native prey”. It could also be stray dogs, garbage … . Moreover, wolves that killed sheep, may also kill native prey. The one class does not exclude the other then. Therefore, we believe it is safer to use the class “known to kill sheep” or “not known to kill sheep”.

To make it clearer, we changed the following starting at line 222: (1) no kill of sheep, no proximity to humans, (2) no kill of sheep, proximity to humans, (3) kill of sheep, no proximity to humans, and (4) kill of sheep, proximity to humans.

To avoid repetition in describing the classes, because they are explained and defined in Material in Methods, we suggest that the references to the classes later on in the text remain as they are.

Though their total sample size is somewhat limited (14 DNA-identified wolves with known parentage) in light of the behavioral categories examined, they show that prey selection and habituation of young, independent wolves largely reflect the behavior of their family in which they were raised. The majority of the wolves killed sheep, and some associated with humans. However, of the few wolves that did not prey on sheep, none of them associated with humans.

I found this paper to be an interesting contribution from a geographic locale that has been underrepresented in the literature on the current topic. However, I feel that the results could be couched with a bit more caution than articulated in the current version of the manuscript

The authors correctly point out that the identity of the wolf for a certain behavior may have been uncertain. As importantly though, is the limited scope that these data represent. Though the statistical relationship between the behaviors of lone wolves relative to their parent pack was statistically significant, the low number of independent samples calls into question the generality of the results (albeit squaring with similar observations from the literature).

We agree that the total sample size of 14 is limited and that we need to take that into account when we discuss the results. The following changes with additional caution have been made:

line: 360

line: 442

line: 449-459

line: 554

line: 599

Moreover, there are possible temporal issues to consider as well. Wolves learn throughout their lives, not just during their adolescence.

We agree, but we focus only on the short period after they leave their parent packs, because the focus of our study was on what the loners have learnt from their parent pack and not later. Nevertheless we extended the paragraph in the discussion starting with:

line 484: In case abundance of familiar wildlife prey is falling, it cannot be ruled out that a naïve wolf starts exploring farm animals like sheep on its own.

We extended this paragraph being a good entry to fulfil remarks of both referees about prey switching, and diversity and abundance of native prey. Therefore the paragraph is extended to discuss roe deer abundance, as this species is the most commonly killed by German wolves. The paragraph continues with:

line 485-498: Roe deer are strongly preferred wild ungulates by German wolves, ….

I see no reason why the current “rabbit hunters” could not in time become sheep killers. Wolves are opportunistic hunters, clearly capable of prey switching as ecological conditions change. Indeed, the paper falls short in providing the reader with information regarding the diversity and abundance of potential native prey in the ecosystem to which these young wolves dispersed. Thus, a fuller description of the broader ecological context experienced by these dispersing wolves could shed additional light onto thesuite of behaviors discussed in this paper.

See also remark above. We agree that the context in the Netherlands should have been described. Therefore, we added in the Introduction section at line 79-92 a description of the Netherlands with elements that relate to the (un)suitability as a wolf habitat. Here we also added descriptions of the ungulate species in the Netherlands that are native wolf prey species in NW Europe.

Round 2

Reviewer 2 Report

I am happy with the revised text of this paper